# Organization of the gravity-sensing system in zebrafish

Zhikai Liu[1], David G. C. Hildebrand [2], Joshua L. Morgan[3], Yizhen Jia[1], Nicholas Slimmon [1] & Martha W. Bagnall [1] ✉

Motor circuits develop in sequence from those governing fast movements to those governing slow. Here we examine whether upstream sensory circuits are organized by similar principles. Using serial-section electron microscopy in larval zebrafish, we generated a complete map of the gravity-sensing (utricular) system spanning from the inner ear to the brainstem. We find that both sensory tuning and developmental sequence are organizing principles of vestibular topography. Patterned rostrocaudal innervation from hair cells to afferents creates an anatomically inferred directional tuning map in the utricular ganglion, forming segregated pathways for rostral and caudal tilt. Furthermore, the mediolateral axis of the ganglion is linked to both developmental sequence and neuronal temporal dynamics. Early-born pathways carrying phasic information preferentially excite fast escape circuits, whereas later-born pathways carrying tonic signals excite slower postural and oculomotor circuits. These results demonstrate that vestibular circuits are organized by tuning direction and dynamics, aligning them with downstream motor circuits and behaviors.

Central neuronal circuits mediate the transformation from sensory inputs to motor outputs. Motor output can be described by both the direction of movement and its temporal dynamics—fast and rapidly fatiguing versus slow and sustained contractions. Fast and slow motor behaviors emerge sequentially, set up by early-born and late-born motor circuits, respectively[1]. In zebrafish, spinal motor neurons differentiate in sequence from those controlling fast movements to those controlling slow[2]. Similarly, premotor neurons in spinal cord and brainstem governing fast escape movements develop earlier than premotor neurons driving slower locomotion[3–5]. Developmental sequence also influences motor circuit connectivity in flies, linking the timeframe of motor circuit formation to temporal dynamics of movement[6,7]. Both fast and slow behaviors can be elicited by a wide range of sensory inputs. It remains unclear whether sensory systems are also organized around principles set up by developmental sequence that align with the speed-dependent motor circuit architecture.

The vestibular system encodes information about head movement in space. Both translational acceleration and orientation with respect to gravity are encoded by hair cells arrayed underneath otoliths in the inner ear. Striolar hair cells carry high-pass signals which are predominantly relayed by irregular-firing vestibular afferents with phasic kinetics, whereas extrastriolar hair cells excite predominantly regular-firing vestibular afferents with tonic or phasic-tonic dynamics[8]. Both types of vestibular afferents relay head movement information into the brainstem, where they excite a variety of central targets. Central vestibular nucleus neurons innervated by vestibular afferents project directly to oculomotor and spinal neurons to drive behaviors[9]. This tight connection from sensory input to motor output suggests that vestibular circuits, like motor and premotor circuits, might be organized around speed-dependent principles. However, it has been difficult to link the patterning of central premotor circuits to hair cell organization because vestibular afferents are not known to be spatially organized by tuning direction or their temporal dynamics.

[1]Dept. of Neuroscience, Washington University in St. Louis, St. Louis, MO, USA. [2]Laboratory of Neural Systems, The Rockefeller University, New York, NY, USA. [3]Dept. of Ophthalmology, Washington University in St. Louis, St. Louis, MO, USA. ✉e-mail: bagnall@wustl.edu

We examined the architecture of the gravity-sensing system in larval zebrafish (*Danio rerio*). Zebrafish begin to exhibit simple vestibular functions, such as postural control and vestibulo-ocular reflexes, as early as 3 days post fertilization[10–12] (dpf). Over the following days and weeks, vestibular behaviors improve and refine[13,14]. The small size of the larval zebrafish brain makes it tractable for reconstruction with serial-section electron microscopy (ssEM) at synaptic resolution. In these animals, only the utricular otolith is responsible for gravity sensation[10,15], and accordingly we reconstructed all utricular hair cells, afferents, and four classes of central neurons receiving utricular inputs.

Directional tuning of utricular hair cells was topographically mapped onto the rostrocaudal axis of the utricular afferent ganglion. Afferents carrying inferred rostral or caudal tilt information excited compensatory central vestibulo-ocular reflex (VOR) circuits to stabilize eye position. Similarly, afferents carrying information about head movement in inferred ipsilateral or contralateral directions excited distinct elements of Mauthner cell escape circuits. Along with these directional maps, we identified cellular signatures of vestibular hair cells and afferents, including cilia length, synapse counts, and myelination, that indicated their developmental sequence. Inferred developmental sequence was mapped onto the mediolateral axis of the utricular ganglion and also correlated with central vestibular targets controlling distinct behaviors. Early-born sensory pathways are preferentially connected to drive early-born fast motor circuits, whereas later-born pathways govern movements mediated by later-born slower motor circuits. Collectively, these data revealed a sensorimotor transformation organized around movement speed, where phasic and tonic vestibular signals are preferentially used to regulate fast and slow movements respectively. Together, the directional and temporal tuning of vestibular circuits pattern the entire vestibulomotor transformation.

## Results

### Imaging the utricular system at synaptic resolution

Gravity and inertial forces are sensed by hair cells in the inner ear. The otolith, or in mammals the otoconial matrix, slides relative to the macula during head tilt or translation to deflect hair cell cilia. Utricular hair cells, which in larval zebrafish serve as the sole gravitational sensors[10,15], synapse onto the peripheral process of utricular afferents (schematic, Fig. 1a). These afferents, whose cell bodies reside within the vestibular ganglion, project axons that bifurcate and synapse in several brainstem nuclei that mediate behaviors like escape, posture, and the vestibulo-ocular reflex (VOR). We obtained an ultrathin section library of the larval zebrafish at 5.5 days post fertilization (dpf), which had originally been imaged at $18.8 \times 18.8 \times 60.0$ nm$^3$ per voxel or $56.4 \times 56.4 \times 60.0$ nm$^3$ per voxel depending on the region[16]. We re-imaged the peripheral and central areas of the right utricular circuit at $4.0 \times 4.0 \times 60.0$ nm$^3$ / voxel (Fig. 1b), sufficient resolution to visualize hair cell cilia (Fig. 1c) and vestibular afferent processes (Fig. 1d). The new images were aligned to the lower-resolution data and used to produce a reconstruction of the gravity-sensing system, including 91 utricular hair cells, 105 ganglion afferents, and ~45 target neurons in the ipsilateral vestibular brainstem (Fig. 1e–g; Supplementary Movie 1).

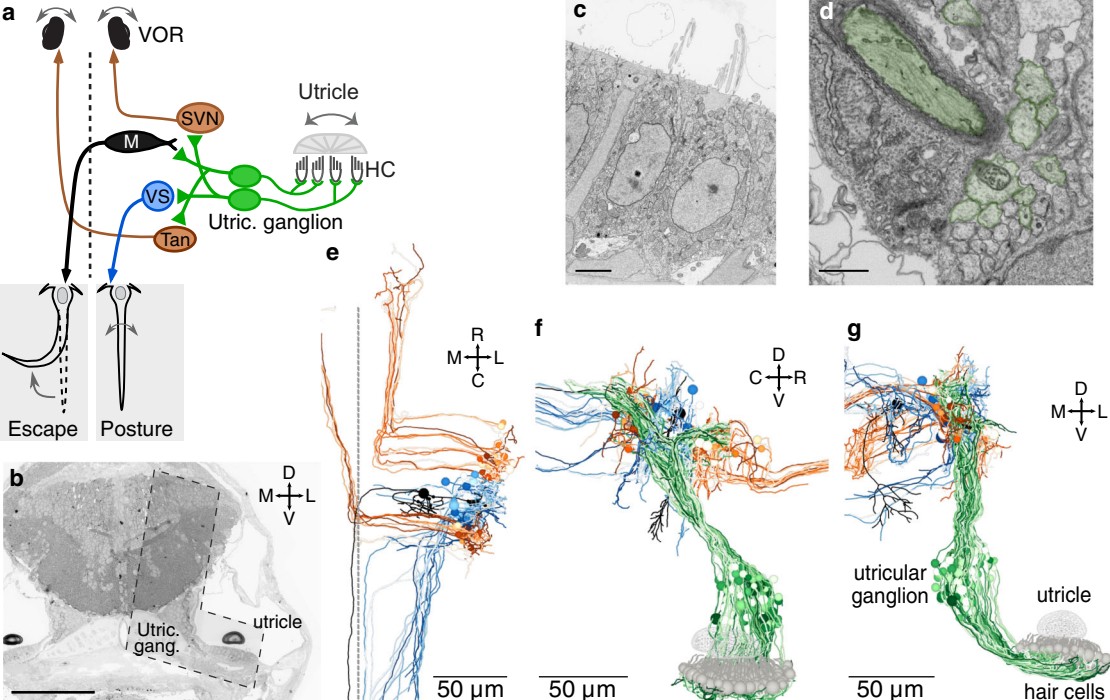

**Fig. 1 | High-resolution serial-section electron microscopy of the gravity-sensing system. a** Schematic of the gravity-sensing system in fish. Hair cells in the utricular macula (HC, gray) are inertial sensors of head tilt and translation, exciting the peripheral process of utricular afferents (green). These afferents, whose cell bodies are located in the utricular ganglion, project to brainstem neurons involved in escape (Mauthner cell, black), posture (vestibulospinal [VS] cell, blue), and oculomotor (VOR) reflexes (superior vestibular nucleus [SVN] and tangential nucleus [Tan], brown). Dashed line indicates midline. **b** Coronal section through the head of a 5.5 dpf zebrafish. The region reimaged at high resolution is visible as an L-shaped territory (dashed outline) covering the right utricle and hair cells, utricular ganglion, and ipsilateral brainstem. The reimaged territory extended across 1757 coronal sections (105 μm in the rostrocaudal axis). Scale bar, 100 μm. **c** Electron micrograph of two hair cells in the utricular macula, with portions of their cilia. Scale bar, 3 μm. Source data for this and all example EM images are provided as links in the Source Data file. **d** Section of the vestibular nerve, peripheral processes. At this developmental stage, some axons are myelinated (pseudocolored dark green) while others are not yet (light green). Scale bar, 1 μm. **e** Horizontal projection of reconstructed brainstem targets (Mauthner, VS, SVN, Tangential) colorized as in **a**. The file to generate this and all other reconstructions in this paper are provided in the Source Data file. **f** Sagittal projection of utricular hair cells, afferents, and brainstem targets, as in **a**. **g** Coronal projection as in **f**.

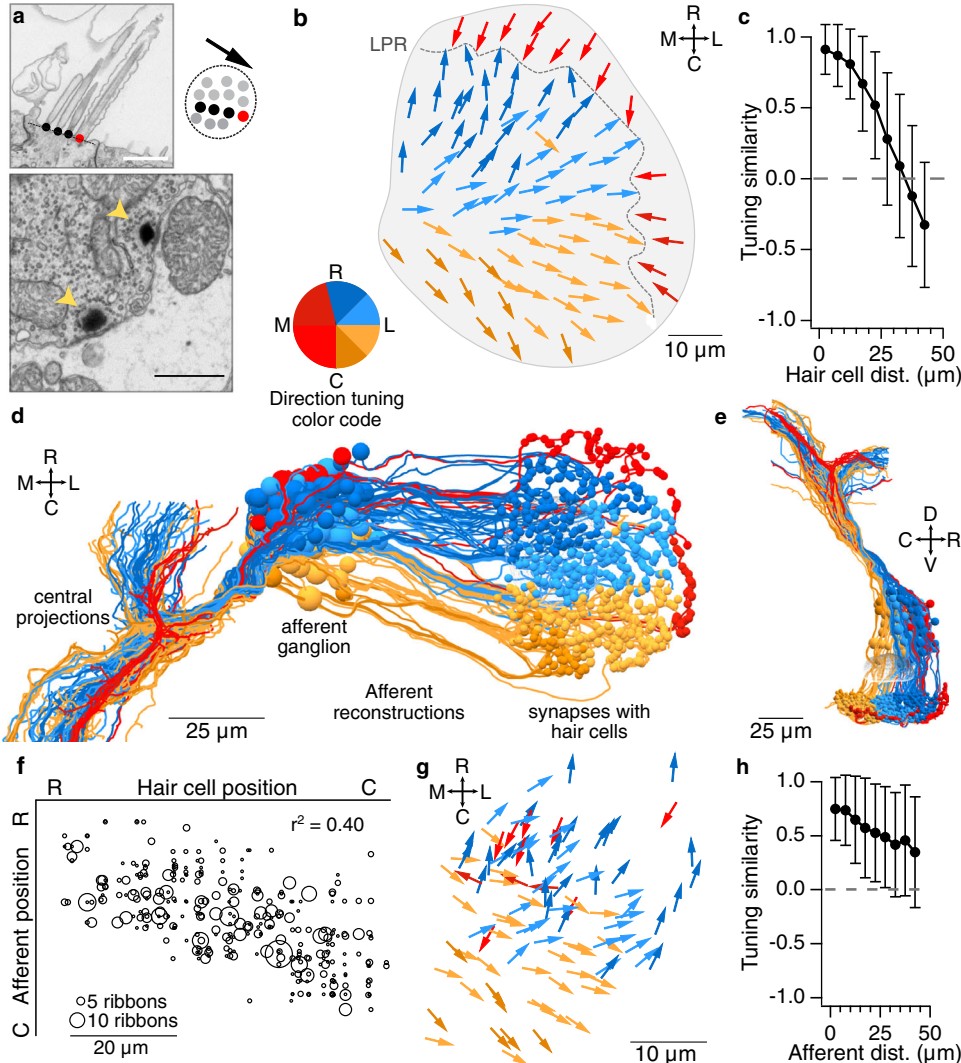

**Fig. 2 | The utricular afferent ganglion is organized in the rostrocaudal axis by directional tuning. a** Top, Electron micrograph of utricular hair cell with stereocilia (black) and kinocilium (red) marked. Right, schematic of tuning vector derived from cilia positions, viewed from above. Bottom, EM image of hair cell synaptic ribbons (arrowheads) apposed to a utricular afferent. Scale bars, 1 μm. **b** Horizontal projection of the utricular macula, showing tuning direction vectors for all 91 hair cells. Dashed line: line of polarity reversal (LPR). Vectors are colorized by directional tuning. Note slight asymmetry in colorization; this was chosen to ensure hair cells from the medial and lateral sides of the LPR are represented in different colors. Here and throughout: R, C = rostral, caudal = nose-down and nose-up pitch, respectively. M, L = medial, lateral = contralateral and ipsilateral roll, respectively. Source data for this and all graphs is provided in the Source Data file. **c** Hair cells located near each other have more similar directional tuning, which falls off sharply within ~30 μm. Data from hair cells medial to the LPR only (*n* = 77 hair cells) and represent mean values ± SD. **d** Horizontal view of reconstructions of all 105

utricular ganglion afferents including somata (larger spheres, left) and their postsynaptic contacts in the utricular macula (smaller spheres, right). Afferents are colorized by inferred direction tuning as in **b**. View is slightly tilted to aid in visualization of ganglion. Afferents with inferred contralateral head tilt tuning (red) form a segregated axon bundle in the brainstem (left). **e** Sagittal view of reconstructions shown in **d**. **f** Correlation of rostrocaudal soma position between synaptically connected utricular hair cells and afferents. Circle size reflects the number of synaptic ribbon connections (range: 1–19). Significance of linear correlation, *t*-test, *p* = 1.3 × 10⁻¹⁰². **g** Horizontal projection of inferred afferent tuning vectors, relative to soma position in the afferent ganglion. Each vector indicates an afferent's tuning direction, calculated by weighting by the number of ribbon inputs it receives from each hair cell. Colors as in **b**. **h** Afferents located close to each other have similar directional tuning, but the relationship is looser than in hair cells (**c**). Data are from afferents innervating the macula medial to the LPR only (*n* = 94 afferents) and represent mean values ± SD.

## Characterization of direction tuning in utricular afferent circuits

We first established the tuning direction of every hair cell in the utricular macula. Hair cells are maximally depolarized by head tilts in the direction of their kinocilium relative to the cluster of stereocilia[8]. We measured directional tuning for each hair cell by drawing a vector from the center of mass of the stereocilia bases to the kinocilium base (schematic, Fig. 2a). Hair cell direction tuning vectors were displayed relative to soma position to yield a sensory map of the entire utricular macula (Fig. 2b), with vectors colorized by their inferred directional tuning. Consistent with prior observations[17,18], the line of polarity reversal (LPR) was found towards the lateral edge. On either side of the

LPR, directional tuning varied smoothly from rostral tilt sensitive to caudal tilt sensitive. As a result, hair cells located near each other tended to have more similar tuning than those far apart (Fig. 2c).

## Organization of the utricular ganglion by directional tuning

We next asked whether this organization of hair cells by directional tuning was reflected in the organization of the utricular afferent ganglion. Despite a rich literature of physiological responses in utricular afferents recorded in their axonal processes[19–21], the bone surrounding this ganglion in adult animals has historically prevented somatic recordings, rendering its sensory topography unknown. To

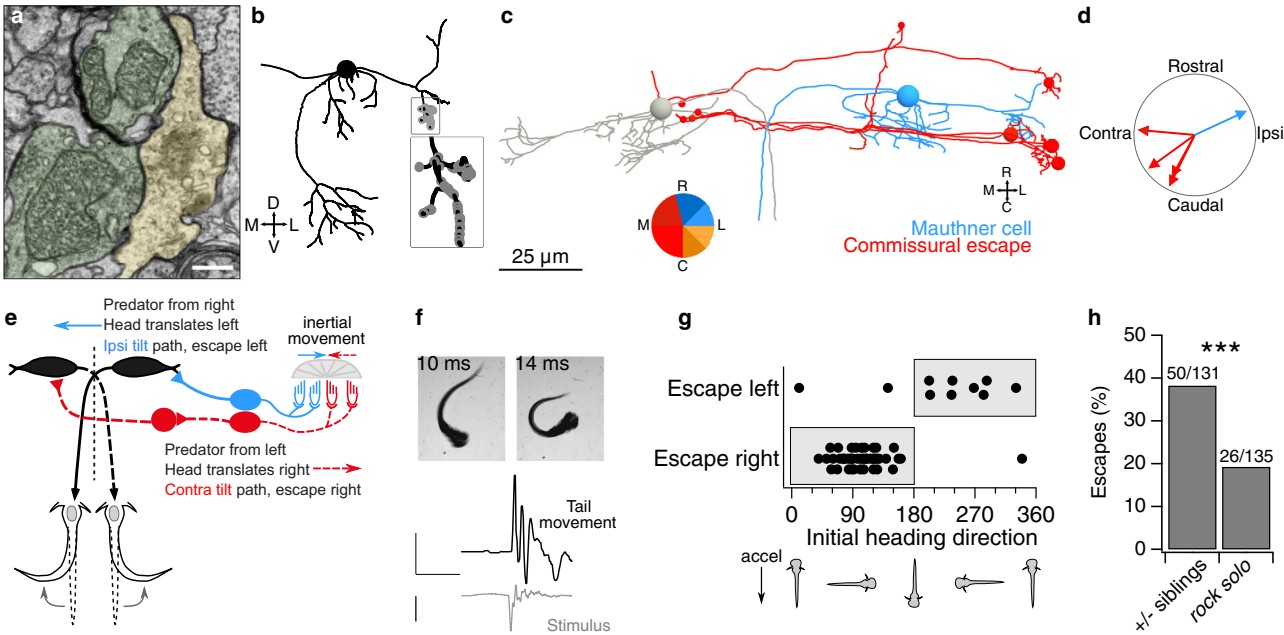

**Fig. 3 | Escape circuits compute head movement direction from utricular input. a** Example micrograph of utricular afferents (pseudocolored green) synapsing onto the lateral Mauthner cell dendrite (yellow). Chemical synapses are recognizable by clustered presynaptic vesicles and synaptic density. The tight apposition between the upper afferent and the Mauthner is likely an electrical synapse. Scale bar, 0.5 μm. **b** Coronal projection of Mauthner cell skeleton reconstruction (black, spherical soma) with utricular afferent input synapses (gray). Inset, expanded view of utricular inputs. **c** Horizontal projection of reconstructions of both Mauthner cells (gray, blue) and four commissural utricular neurons (red). Commissural utricular neurons make synaptic contacts on the contralateral Mauthner (small red circles). Colors indicate inferred directional tuning. **d** Polar plot of inferred direction tuning of utricular input to Mauthner cell and four commissural escape neurons. Directional tuning is indicated in the context of head tilt. Note that as inertial sensors, otoliths are equally sensitive to head translation in the opposite direction (*e.g.*, ipsilateral head tilt and contralateral translation are indistinguishable). **e** Schematic of predicted Mauthner cell computation of head translation. A predator approaching from the right will cause a head deflection to the left. Deflection of the utricular otolith by inertia (blue

arrow) would depolarize the ipsilateral tilt / contralateral translation pathway (blue; medial to the LPR). These utricular afferents excite the ipsilateral Mauthner cell, promoting an escape movement to the left. Commissural escape neurons, in contrast, will respond to rightward head movements (red hair cells and dashed lines) and are predicted to activate the contralateral Mauthner cell, promoting escapes to the right. **f** Example of behavioral response in a free-swimming larva subjected to rapid translation. High-speed videography captures the onset of escape and characteristic C-bend. Bottom, quantification of tail angle and translational stimulus. Scale bars, 1 rad, 100 ms, and 1 *g*. **g** Escape responses to a unidirectional translational stimulus are plotted relative to the larval heading angle at the start of the stimulus. As predicted, larvae accelerated to the right (heading direction 0–180°) escape to the right, whereas larvae accelerated to the left escape left. Escapes only occurred in the "incorrect" direction when animals were accelerated in predominantly rostral or caudal directions. **h** With both types of stimulus, utricle-deficient *rock solo* larvae escaped at approximately half the rate as their heterozygous siblings. N = 52 sibling and 55 −/− fish. Chi-squared test, p = 0.0006 (1 degree of freedom, chi-squared value = 11.65, 266 total observations).

create a map of utricular ganglion topography, we reconstructed all processes postsynaptic to the hair cell ribbon synapses (Fig. 2a, bottom), which are by definition utricular afferents, back to their soma locations in the utricular ganglion. Out of 944 ribbon synapses, 929 (98.4%) were apposed to afferent processes that could be followed out of the macula. A total of 105 utricular afferents were reconstructed, with an average of 3.0 ± 1.5 discrete hair cells contacting each afferent, and 3.4 ± 1.4 afferents contacting each hair cell (mean ± SD; Supplementary Data 1). In a 3D reconstruction of all afferents, there was a visible rostrocaudal gradient to their organization (Fig. 2d; note that this reconstruction is tilted slightly relative to a true horizontal view so that the afferent ganglion is visible beneath the central axon projections).

Afferents whose somata were rostrally located within the ganglion were innervated by hair cells in the rostral utricular macula, whereas afferents in the caudal portion of the ganglion were innervated by hair cells in the caudal macula. Consistent with previous reports[22], each afferent received input from hair cells exclusively on one side or the other of the LPR. To infer directional tuning of each utricular afferent, we weighted its convergent hair cell vectors by their number of ribbon synapses. Here and throughout the manuscript, this anatomically inferred directional tuning will be referred to more simply as directional tuning, but we note that actual directional tuning might differ

based on variations in synaptic weight, process morphology, and other biological variables that cannot be assessed with EM.

The resulting map revealed a sensory tuning topography in the utricular ganglion, with rostrally located afferents encoding rostral head tilts and caudally located afferents encoding caudal head tilts (Fig. 2d, e; Supplementary Movie 2). Quantifying hair cell to afferent connectivity revealed a strong correlation between the rostrocaudal position of each afferent soma and the hair cells that innervate it (Fig. 2f). However, unlike in the hair cell macula (Fig. 2b), afferents innervating hair cells lateral to the LPR are intermingled with those innervating hair cells medial to the LPR (Fig. 2g, note red arrows), both in this horizontal view and in sagittal and coronal views (Supplementary Fig. 2a, b). Nonetheless, these afferents with inferred contralateral tilt tuning formed a segregated axon bundle in the brainstem (Fig. 2d; Supplementary Movie 3). Afferent somata located near each other typically had more similar tuning than afferents that were further apart (Fig. 2h), although the strength of the relationship is less pronounced than in the hair cells themselves (cf. Figure 2c).

## Central organization of directional tuning
We next asked how directional tuning was organized centrally in three utricular afferent target populations: the Mauthner escape circuit, the VOR circuit, and the vestibulospinal (VS) postural circuit. Our high-

resolution reimaged territory in the brainstem allowed us to identify the postsynaptic targets of utricular afferents. These central brainstem targets were then reconstructed and characterized based on soma location and axon projection. Although most central vestibular neurons project axons outside the vestibular brainstem, into lower-resolution regions that were not reimaged, they could be reconstructed over long distances if their axons were myelinated[16]. Within these criteria, we characterized utricular input to the ipsilateral Mauthner cell, 4 commissural escape neurons projecting to the contralateral Mauthner cell, 23 VOR neurons, and 19 VS neurons. We note that additional commissural, VOR, and VS neurons presumably exist with insufficiently myelinated axons for adequate reconstruction, and therefore the set sampled here is likely biased towards earlier-born neurons. Additional neurons of less certain identity are described elsewhere[23].

The Mauthner cells are a specialized pair of reticulospinal neurons that trigger fast escape movements in response to multiple types of sensory input. Mauthner cells develop by 8 h after fertilization, excite primary (fast, early-born) spinal motor neurons, and drive fast escape movements by 1 dpf[24–26]. We identified and reconstructed utricular inputs both onto the ipsilateral Mauthner cell and onto brainstem neurons whose axons cross the midline and appear to synapse onto the contralateral Mauthner cell. In total, 18 utricular afferents contacted the Mauthner cell lateral dendrite (example, Fig. 3a) with a total of 52 synaptic contacts. All utricular afferent synapses were clustered together tightly on ventral sub-branches of the Mauthner cell lateral dendrite (Fig. 3b), as reported[27]. In addition, seven utricular afferents synapsed on a commissurally projecting neuron population that contacted the contralateral Mauthner (Fig. 3c), which we term "commissural escape" neurons (Supplementary Movie 3).

Interestingly, no utricular afferents diverged to contact both the ipsilateral Mauthner cell and this commissural escape population. Utricular afferents presynaptic to the Mauthner cell were innervated by hair cells in the medial portion of the macula, whereas afferents that excited the commissural-projecting neurons were innervated by hair cells in the lateral portion of the macula, on the far side of the LPR. The inferred tuning vectors of all afferents contacting the Mauthner cell were averaged to yield an inferred tuning to tilt in the ipsilateral and rostral direction (Fig. 3c, d). Tuning vectors of commissural escape neurons pointed in the opposite direction, contralaterally and caudally (Fig. 3c, d; average difference between Mauthner and commissural neuron inferred tuning, 196°). Therefore, these commissural escape neurons are presumably similarly tuned to inertial movements as the contralateral Mauthner cell that they contact.

Vestibular stimuli can elicit locomotion[28,29], but escape behaviors have primarily been evoked with auditory or mixed auditory-vestibular stimuli that do not allow isolation of the vestibular component[30,31]. We reasoned that the utricular afferents exciting the Mauthner cell are tuned to report ipsilateral head tilts and therefore contralateral translational movements, with the predicted consequence that a translational movement to the left should trigger escape bends to the left (Fig. 3e). This type of circuit would be useful to detect predator movement towards the zebrafish. While the water flow of a predator bow wave can be detected by lateral line circuits[32], there is little known about whether the head deflection itself can also elicit escapes. To test this prediction, we delivered a large amplitude translational stimulus optimized for speed (~10 ms, >1 g) to freely moving zebrafish larvae while recording behavior at 508 frames/s. This stimulus evoked a classic short-latency escape response in 35% of trials (Fig. 3f; 28 escapes in 81 trials from 27 animals; escapes defined as C-bends within 10 ms of peak acceleration[30]). Animals escaped in the direction predicted by the circuit in Fig. 3e on 64% of trials (18/28). However, due to sled limitations, this stimulus involved bidirectional movement to maximize the accelerative force. We therefore repeated this experiment with a second stimulus that was slightly slower but optimized to

be unidirectional (~20 ms, 0.8 g). This stimulus elicited a similar frequency of escapes (28%: 56 escapes in 202 trials from 105 animals). Notably, successful escapes were strongly directionally biased: animals turned in the direction of peak acceleration on 95% (53/56) of escapes (Fig. 3g, shaded regions). Turns in the "wrong" direction occurred on trials where animals were accelerated rostrally or caudally (Fig. 3g). To test for utricular dependence, both stimuli were presented to the utricle-deficient *rock solo* fish line, an otogelin mutant[10,33]. Utricle-deficient animals escaped about half as often as their sibling controls (Fig. 3h). Because experiments were carried out under infrared light, the remaining escapes were likely triggered by the lateral line system[32]. Based on these anatomical and behavioral results, we conclude that this utricular-activated escape circuit allows for computation of the direction of head deflections, such as occur during predator approach.

Next, we evaluated directional tuning in three vestibular nuclei. VOR neurons of the superior vestibular nuclei (SVN) and tangential nuclei collectively govern vertical and torsional eye movements. We reconstructed 12 neurons in the SVN, which inhibits the ipsilateral trochlear and oculomotor nuclei[9], and 11 neurons of the tangential nucleus[12], which excites contralateral trochlear and oculomotor neurons (Fig. 4a, b). Of these VOR neurons, 22/23 had inferred tuning in the ipsilateral rostral or caudal directions (insets, Fig. 4a, b), consistent with their well-described roles in the VOR[9]. This inferred tuning also aligns with the angles of the anterior and posterior semicircular canals, which contribute to rotational VOR behaviors later in development[9]. Notably, the SVN and tangential neurons tuned for rostral vs caudal tilt also projected axons with largely distinct trajectories. SVN axons with inferred rostral tilt encoding traveled more laterally than those encoding caudal tilt (boxed inset, Fig. 4a), and presumably inhibit the motor neurons that drive downward eye rotation via the inferior rectus and superior oblique muscles[34]. Tangential axons with inferred rostral tilt encoding traveled ventrally and are presumed to activate the activate the eyes-up pathway through motor neurons that control the superior rectus and inferior oblique, whereas those with inferred caudal tilt encoding traveled dorsally where they likely activate the eyes-down pathway through motor neurons of the inferior rectus and superior oblique[35–37] (Fig. 4b; Supplementary Movie 4).

Similarly, we characterized the tuning of afferent inputs to the VS population, which is involved in postural control (Fig. 4c)[38]. We extended our previously reported connectivity from myelinated utricular afferents[39] to include the unmyelinated afferents. The 19 VS neurons received input from a total of 61 afferents. In contrast to VOR nuclei, VS neurons primarily received input from more rostral tilt sensitive afferents (Fig. 4c). VS neurons were typically contacted by a greater number of distinct utricular afferents than VOR pathway neurons were (medians: 8 afferents per VS neuron, 25th–75th %ile, 3.5–15.5; 4 afferents per VOR neuron, 25th –75th %ile, 3–7; Wilcoxon–Mann–Whitney between the number of distinct utricular afferents contacting VS and VOR neurons, $p = 0.036$). Thus VS neurons are contacted by a large number of predominantly rostral tilt sensitive utricular afferents.

Little is known about subcellular organization of vestibular afferent input onto central neurons. The concentration of utricular inputs onto a portion of dendrite in the Mauthner cell (Fig. 3b) led us to use the high resolution of EM to examine whether utricular inputs were similarly concentrated in VOR and VS neurons. Dendrograms of utricular afferent input to VOR and VS neurons revealed that most utricular input is not evenly distributed across the dendritic arbor but instead arrives on a small subset of branches (examples, Fig. 4d). To quantify the concentration of synapses, we carried out Monte Carlo simulations of synaptic distribution with either randomly distributed synapses or locations weighted by distance from afferents (see Methods). We found that utricular afferent synapses were located more closely to each other than expected by random chance, likely related to the limited spatial range over which utricular afferents interact with vestibular dendrites (Fig. 4e, Supplementary Fig. 5). This

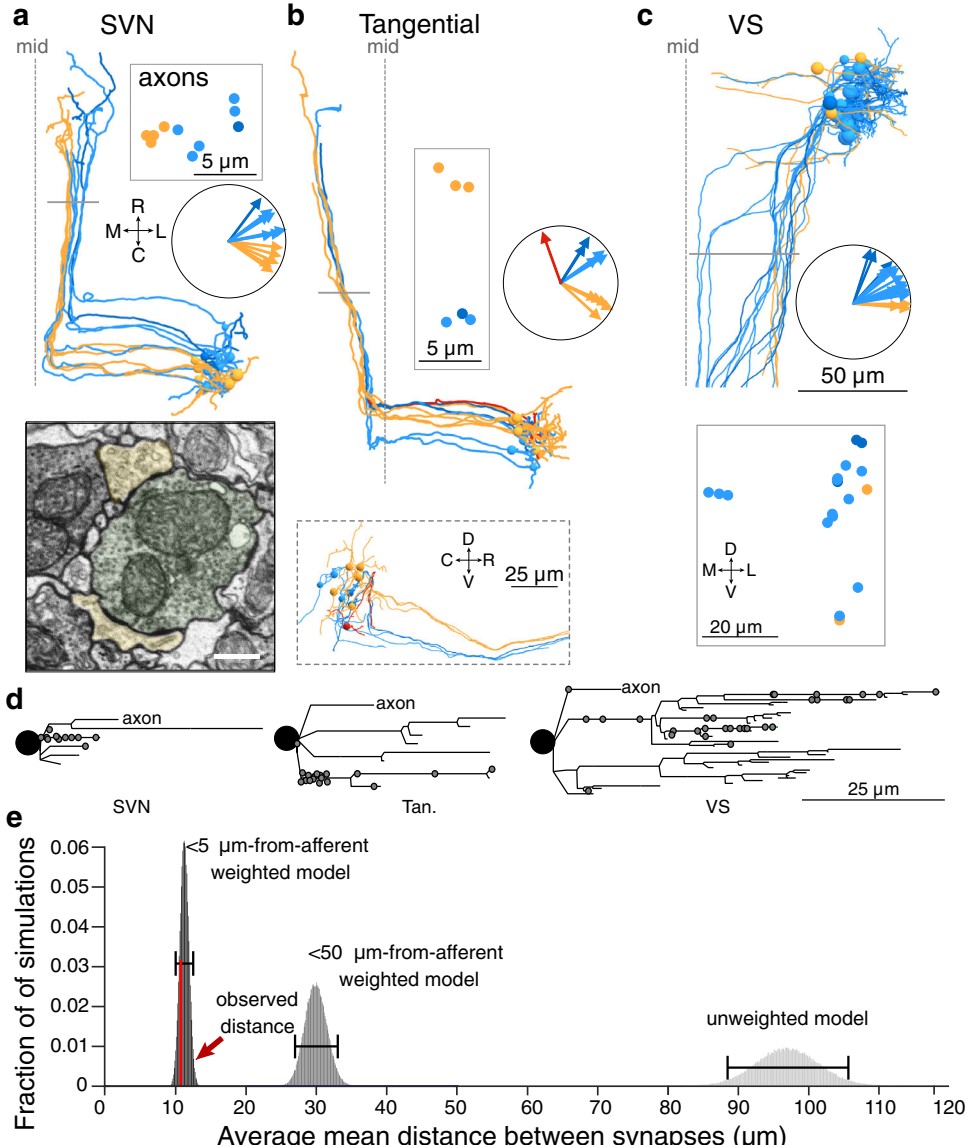

**Fig. 4 | Structure and tuning of central VOR and VS utricular targets.**
**a** Horizontal projection of 12 SVN reconstructed neurons. Neurons are colorized based on their inferred directional tuning. Polar plot inset shows the directional tuning vectors of each neuron. Gray bar indicates location of coronal plane where axon trajectories are shown in boxed inset. Scale for reconstructions as in **c**. Electron micrograph (bottom) shows a utricular afferent (pseudocolored green) contacting two dendrites of SVN neurons (yellow). Scale bar, 500 nm. **b** As in **a**, for 11 tangential nucleus reconstructed neurons. Sagittal projection inset (bottom) shows the divergence of tangential neuron axons in the dorsoventral axis. **c** As in **a** but for 19 VS neurons. **d** Dendrograms of three example neurons from the SVN, tangential,

and VS populations. The soma is represented by a black circle and each dendrite is represented by lines. Gray circles indicate synaptic inputs from utricular afferents, which appear disproportionately concentrated on a small number of dendrites. Axons are truncated for purposes of scale. These and all dendritic arbors are provided in the Source Data file. **e** Quantification of synaptic clustering, measured as distances between synapses, using three Monte Carlo models compared to actual data. Synaptic locations were modeled as randomly distributed across the arbor (unweighted model), preferentially weighted within 50 μm of afferent axons, or preferentially weighted within 5 μm of afferent axons. The observed level of synaptic clustering is shown in red. See Methods for detailed description.

result indicates that branch-specific computation may occur in vestibular nucleus neurons, perhaps in conjunction with localized cerebellar input[40].

These analyses revealed a rostrocaudal map of direction tuning in the utricular afferent ganglion, as well as patterns of direction tuning in central brainstem targets of the utricular afferents. We next considered the organization of temporal kinetics arising from distinct types of hair cells.

**Temporal dynamics and developmental sequence of utricular hair cells**
In addition to their directional tuning, hair cells are characterized as striolar or extrastriolar, based on morphological differences in soma

shape and ciliary lengths[41–44]. Striolar hair cells typically drive phasic, adapting, irregular-firing afferents with high-pass sensitivity, whereas extrastriolar hair cells drive tonic or tonic-phasic regular-firing afferents that are less dependent on stimulus frequency[8,19,43,45–47]. Fish and frogs do not express the classical Type I striolar cell shape seen in amniotes, but striolar hair cells can still be recognized based on the length of the kinocilium relative to the tallest stereocilium[44,48]. In the adult zebrafish utricle, striolar hair cells are estimated to have kinocilia and tallest stereocilium lengths around 5 μm, whereas extrastriolar hair cells have a kinocilium estimated at 6–8 μm and tallest stereocilium 2–3 μm[17]. We reconstructed the kinocilium and tallest stereocilium of each hair cell (Fig. 5a) and plotted the relationship between their lengths. In one group of hair cells, both kinocilium and stereocilium

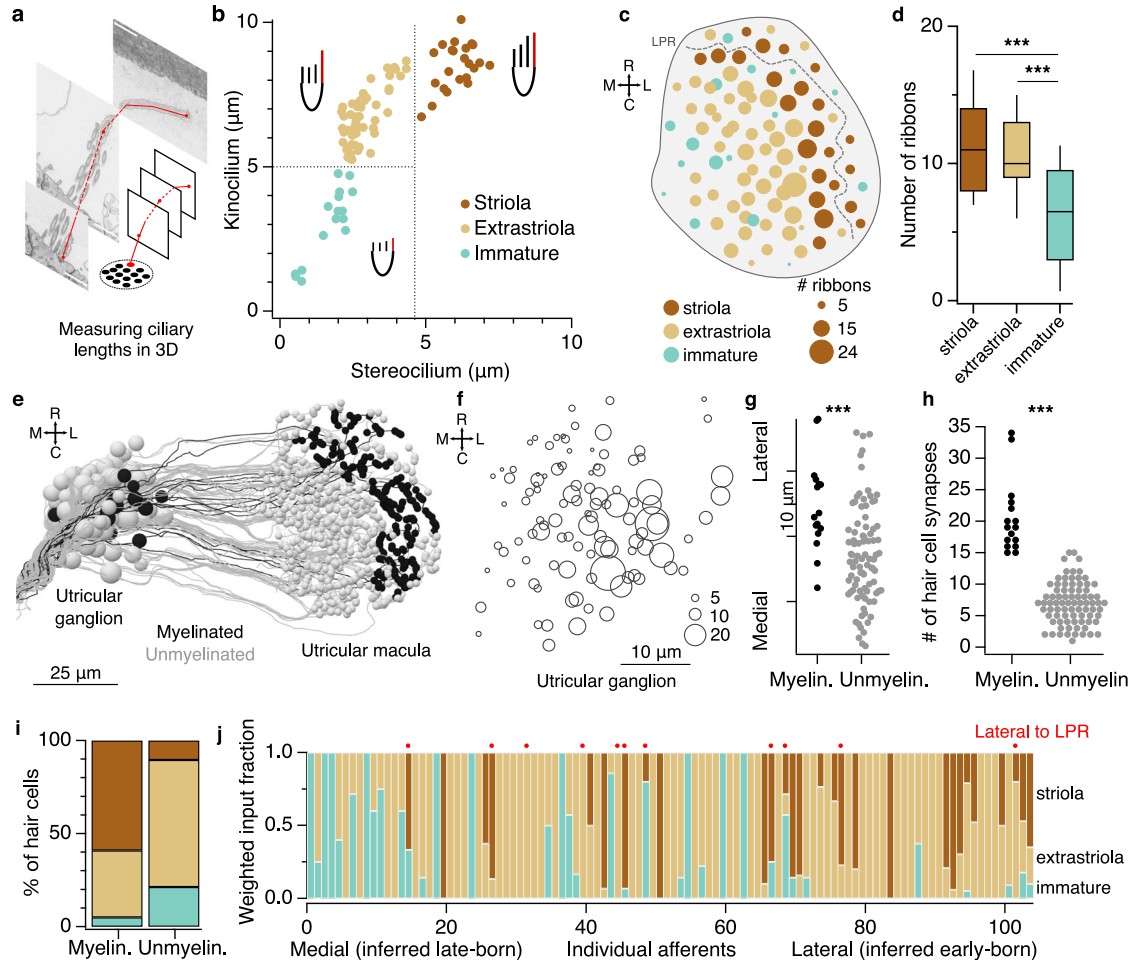

**Fig. 5 | The utricular afferent ganglion is organized in the mediolateral axis by developmental sequence and temporal dynamics. a** Representation of kinocilium reconstruction across successive images to obtain the total length. The neighboring tallest stereocilium was also reconstructed (not schematized). **b** Plot of length of the kinocilium vs. the tallest stereocilium for all 91 hair cells. Striolar hair cells are identified by their long stereocilia and low K/S ratios, whereas extrastriolar hair cells have higher K/S ratios. Hair cells characterized as immature have short kinocilia and stereocilia. **c** Horizontal projection of the utricular macula, showing number of synaptic ribbons in each hair cell. Circle diameter reflects synaptic ribbon count; hair cells with larger numbers of ribbons tend to be located more centrally. See also Supplementary Fig. 1. **d** Quantification of synaptic ribbon counts across hair cell categories. Box plot represents medians ± 25%ile; whiskers indicate 10-90%iles. Immature hair cells form fewer ribbon synapses than striolar or extrastriolar hair cells. $N = 23$ striolar, 52 extrastriolar, and 16 immature hair cells. Significance was tested with ANOVA ($p = 4.8 \times 10^{-5}$) and then pairwise with Wilcoxon−Mann−Whitney (striola−extrastriolar, $p = 0.60$; striola−immature, $p = 4.3 \times 10^{-4}$; extrastriolar−immature, $p = 1.3 \times 10^{-4}$). **e** Horizontal projection

of all utricular afferents, colorized by whether they are myelinated or not. **f** Horizontal projection map of all utricular ganglion somata by position; circle diameter reflects the number of hair cell ribbon synaptic inputs that each one receives. **g** Myelinated afferent somata are located more laterally in the utricular ganglion. Each dot represents one afferent soma. Wilcoxon−Mann−Whitney, $p = 2.9 \times 10^{-4}$. **h** Myelinated afferents are contacted by significantly more hair cell ribbons than unmyelinated afferents. Wilcoxon−Mann−Whitney, $p = 1.8 \times 10^{-10}$. See also Supplementary Fig. 3c. **i** Myelinated afferents receive the majority of their input from striolar hair cells, whereas unmyelinated afferents receive most of their input from extrastriolar and developing hair cells. The distributions are significantly different (chi-squared test, $p < 1 \times 10^{-10}$). See also Supplementary Fig 3d, e. **j** The weighted fraction of inputs each utricular afferent receives from the different hair cell classes. Afferents are ordered based on soma position from medial to lateral. Red dots identify afferents receiving input from hair cells lateral to the LPR (contralateral tilt sensitive). Laterally positioned afferents typically have predominantly striolar inputs, whereas medially positioned afferents have predominantly extrastriolar or immature inputs.

were >4.8 μm and the kinocilium to stereocilium length (K/S) ratio ranged from 1.1–1.7; these were identified as striolar (Fig. 5b). In a second group, the kinocilium was >4.8 μm but the tallest stereocilium was <5 μm, yielding K/S ratios from 1.75–3.3, which we identified as extrastriolar. These ciliary lengths are close to those in the adult zebrafish, suggesting that these are mostly mature hair cells. Finally, at the tail end of the ciliary length distributions, we identified hair cells in which the kinocilium was <5 μm; we classified these as immature, though they presumably exist on a continuum with established hair cells (Fig. 5b). Some of these immature hair cells were also characterized by less cytoplasm, with fewer mitochondria and vesicles, consistent with this identification.

We marked the position of each cell by its kinocilium and found that striolar hair cells straddled the LPR while extrastriolar hair cells populated the rest of the macula, as expected[44,49] (Fig. 5c). Utricular hair cells differentiate in a temporal sequence, with roughly radial development from the center or striola to the periphery[50–54]. We wondered whether hair cell ribbon counts, which generally increase during development[55], could establish additional signatures of this ongoing sequence of differentiation. Counts of the total number of ribbon synapses (example, Fig. 2a) showed that hair cells located centrally in the macula had the highest number of ribbons per cell, with fewer ribbons found in peripheral hair cells (Fig. 5c). Accordingly, striolar hair cells overall had the largest number of ribbons, followed

by extrastriolar and immature hair cells (Fig. 5d). Even within the extrastriolar population, hair cells with more ribbons were located more centrally, and hair cells with fewer more peripherally, consistent with a radial growth pattern. Thus, both ciliary lengths and ribbon synapse number are consistent with a radial pattern of growth in the macula, providing proxies for developmental sequence.

## Organization of the utricular ganglion by developmental sequence

We next asked whether similar proxies for developmental sequence could be identified in the utricular afferent ganglion neurons. Myelination is responsive to neuronal activity[56,57], and evidence suggests that it initiates first in early-born neurons[58,59]. Of utricular afferents, 16 of 105 (15.2%) were myelinated throughout most of their central and peripheral processes; the remaining afferents were mostly or entirely unmyelinated (examples, Fig. 1d), similar to early developmental stages in other animals[60–62]. Because the entire vestibular nerve is myelinated by adulthood[22,60], the myelinated afferents we observed are likely to be early-born, instead of a specialized category of afferents. Consistent with this idea, we found that afferent somata that are myelinated (Fig. 5e) and that receive the highest numbers of ribbon contacts from hair cells (Fig. 5f) occupy the most lateral edge of the nascent ganglion, the site of the earliest-born vestibular afferents[63–65]. Later-born afferents are added more medially, forming a half-shell around the earliest-born somata[63–65]. These observations are supported by quantification of mediolateral position (Fig. 5g) and the ribbon synapse count (Fig. 5h) in myelinated versus unmyelinated afferents. Thus, afferent myelination and soma position are proxies for developmental sequence.

Do early born hair cells preferentially connect with early born afferents? We found that myelinated afferents received over half of their input from striolar hair cells, and little input from immature hair cells (Fig. 5i). This was true whether quantified by the number of distinct hair cells providing input or the total number of synaptic ribbon connections (Supplementary Fig. 3d, e). In contrast, unmyelinated afferents predominantly received input from extrastriolar hair cells, and had a higher proportion of input from immature hair cells (Fig. 5i). At the level of individual neurons, afferents with laterally located somata tended to receive input from striolar hair cells, whereas medially located somata were the most likely to receive immature hair cell input (Fig. 5j).

We conclude from these results that early-born afferents, marked by early myelination, preferentially receive input from early-born, predominantly striolar hair cells, and occupy a lateral position in the ganglion. Later-born afferents contact later-born, predominantly extrastriolar hair cells, and occupy progressively more medial positions, forming a half-shell around the early-born afferents. At the same time, directional tuning is preserved in the rostrocaudal axis, with rostrally located afferents contacting rostrally located hair cells and vice versa (Fig. 2). Thus, the utricular ganglion is organized by directional tuning in the rostrocaudal axis and developmental sequence in the mediolateral axis. Moreover, because inferred afferent temporal dynamics are aligned with developmental order, the ganglion mediolateral axis is also organized from phasic, striolar-dominated afferents at the lateral edge to tonic, extrastriolar-dominated afferents more medially. Therefore, both the spatial and temporal tuning of vestibular signals are topographically organized in the utricular ganglion, though more loosely than at the macula.

## Organization of central circuits by developmental sequence

Building on our observation that developmental sequence is an organizing principle of the utricular afferent ganglion and its connectivity with peripheral hair cells, we asked whether developmental sequence might also be linked to central connectivity. The downstream targets of vestibular afferents become functional at different times in development: Mauthner cells are the first to form connections and drive escape behavior, whereas VOR and VS neurons driving eye and postural movements are born later[38,66]. Does the developmental sequence of utricular afferents predict their patterns of connectivity to brainstem targets? We mapped the hair cells contributing input to afferents that drove escape, VOR, or VS neurons (Fig. 6a). Central escape neurons (the Mauthner cell and the commissural escape neurons) received most of their input from afferents contacting striolar hair cells. In contrast, afferents exciting VOR and VS neurons received input from a much broader territory of the hair cell macula (Fig. 6a). We quantified hair cell input to these pathways by weighting each central synaptic contact by its afferent's fraction of input from striolar, extrastriolar, or immature hair cells. Around two-thirds of utricular inputs to the escape circuits arose from striolar pathways, whereas VOR and VS neurons received a much larger portion of their inputs from extrastriolar regions (Fig. 6b). Similarly, the Mauthner cell and commissural escape neurons also received a higher proportion of their utricular inputs from myelinated afferents, in comparison to VS and VOR populations (Fig. 6a, c). Consistent with the interpretation that myelinated afferents are early-born and more mature, they also diverge to more postsynaptic target nuclei (Myelinated afferents diverging to Mauthner, SVN, tangential, and VS neurons: 9/16; to three targets, 3/16; to two targets, 1/16. Unmyelinated afferents diverging to all four targets, 0/90; to three targets, 9/90; to two targets, 25/90. Supplementary Data 2.) Therefore, early-born hair cells preferentially signal via early-born afferents to early-born brainstem populations, which in turn drive early-born spinal motor circuits for escape. Similarly, later-born hair cells excite later-born circuit elements for posture and oculomotor control.

Developmental sequence may play a significant role not just across but also within neuronal populations. We found that three VS neurons whose axons travel more medially in the brainstem before joining the rest (VS$_{med}$) receive more input from afferents carrying immature hair cell information, whereas four VS neurons whose axons travel more ventrally (VS$_{vent}$) receive more input from afferents carrying striolar information, as compared to other VS neurons (Fig. 6d). Based on these inputs, VS subpopulations with distinct axonal trajectories are predicted to exhibit different temporal dynamics, with more phasic information carried by the VS$_{vent}$ and more tonic by VS$_{med}$. Though the postsynaptic targets of these subpopulations of VS neurons are not known, this result may help identify circuits underlying later refinement of postural control[14].

Collectively, our anatomical analyses demonstrate that directional tuning and developmental sequence pattern the entire vestibulomotor transformation (Fig. 7). The rostrocaudal axis of hair cell organization is largely preserved in the afferent ganglion, leading to distinct pathways that drive responses to pitch. Afferents encoding contralateral tilt are intermingled at the level of the ganglion but form spatially segregated pathways that underlie head direction computation in the escape circuit (Fig. 7a). At the same time, early-born hair cells contact early-born utricular afferents, which in turn preferentially drive escape behaviors mediated by fast, early-developing motor circuits. Later-born sensory pathways support postural and oculomotor behaviors mediated by a mixture of fast and slow muscles (Fig. 7b). These results also demonstrate that striolar signals, which are carried mostly by irregular-firing afferents with predicted high-pass, phasic encoding properties, establish connections with circuits driving rapid-onset reflexes via fast motor pools.

## Discussion

Here we show that the utricular afferent ganglion, which carries gravity sensation into the brain, is organized along two axes: a rostrocaudal axis for directional tuning, and a mediolateral axis for development. Afferents with different directional tuning excite distinct brainstem populations. Further, we demonstrate that early-born afferents

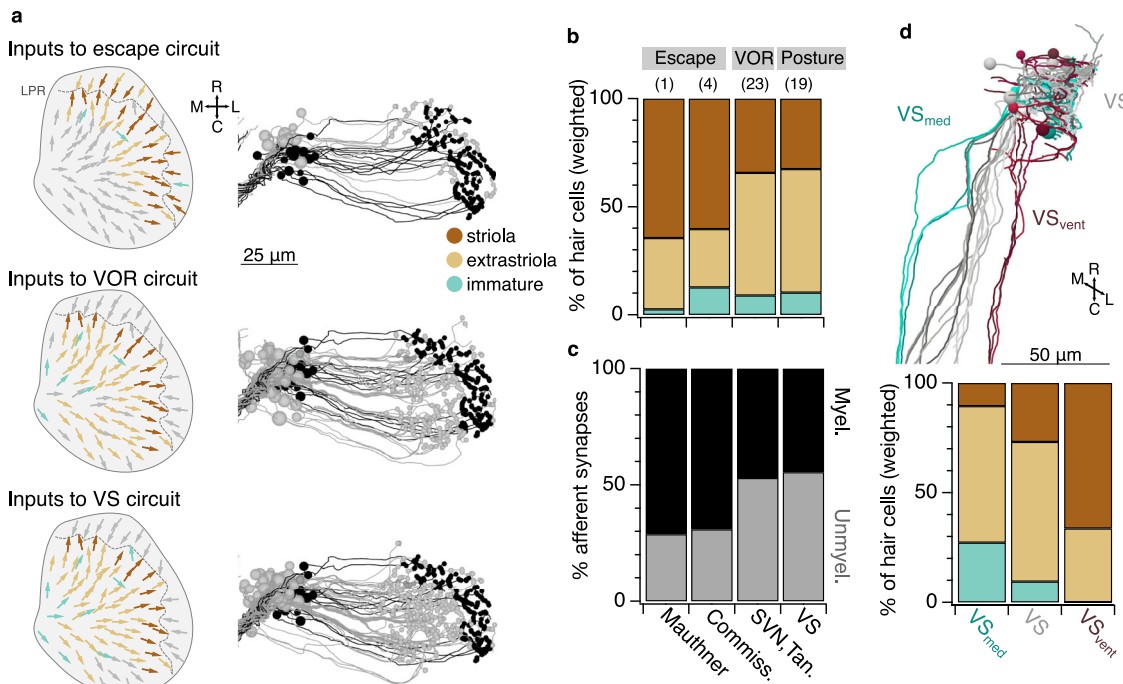

**Fig. 6 | Early developing afferent pathways with fast kinetics preferentially drive early developing central neurons for fast escapes. a** Hair cells in the utricular macula that excite afferents connected to Mauthner escape, VOR, or VS circuits (top, middle, and bottom). Hair cells are colorized if they contribute input to a pathway or gray if they do not. Right, afferents connected to escape, VOR, or VS circuits, colorized by their myelination status (black: myelinated; gray, unmyelinated). **b** Quantification of the contribution of striolar, extrastriolar, and immature hair cells to these central pathways. Striolar hair cells preferentially drive rapid escape circuits, both via the direct afferent input to the ipsilateral Mauthner and the afferent input to the commissural escape pathway, whereas VOR and VS circuits receive input from a mixture of pathways, with extrastriolar inputs dominating. Numbers in parentheses indicate the number of central neurons in each category. Wilcoxon test, striolar contribution to escape vs non-escape neurons, $p = 0.041$.

**c** Quantification of the contribution of synapses from myelinated and unmyelinated afferents to each central pathway. Afferents driving escape pathways are largely myelinated at this age, whereas afferents driving VOR and postural pathways are more mixed. Chi-square test for all groups, $p = 6.0 \times 10^{-8}$. Follow-up chi-square: VOR vs escape, $1.4 \times 10^{-4}$; VS vs escape, $1.7 \times 10^{-8}$. **d** Subsets of VS neurons, identified by axon trajectories, are predicted to exhibit different temporal kinetics. VS neurons with axons that approach the midline before descending (VS$_{med}$, greens) receive mostly extrastriolar (tonic) input with a large contribution from immature hair cells, whereas VS neurons with ventral axon trajectories (VS$_{vent}$, dark reds) receive mostly striolar (phasic) input and none from immature pathways. The skeleton reconstruction is projected at a mixed horizontal/sagittal angle to facilitate visualization of these groups. See also representation of axon trajectories in Fig. 4c and Supplementary Fig. 4.

preferentially receive information from early-born and striolar hair cells, yielding a gradient from phasic to tonic signals in the ganglion. This developmental organization aligns vestibular signals with downstream motor circuits. Brainstem neurons governing the fastest motor circuits, which underlie early-developing escape behaviors, are excited mostly by utricular afferents with inferred phasic firing and early development. Oculomotor and postural circuits, which drive slower motor neurons, are excited preferentially by afferents with inferred tonic firing and late development. Collectively, these results demonstrate that the vestibular circuit is organized by both directional tuning and temporal dynamics to mediate transformation into motor outputs.

We demonstrate here that the utricular afferent ganglion is patterned both by directional sensitivity in the rostrocaudal axis, and by developmental sequence, correlating to temporal dynamics, in the mediolateral axis. As far as we are aware, this is the first demonstration of any topography in the vestibular afferent ganglia. A similar mediolateral gradient has been described for the nearby lateral line afferent ganglion, where early-born afferents are positioned more laterally than later-developing afferents[67]. Early-born lateral line afferents also exhibit larger soma size, lower excitability, and more dorsal central projections than later-born afferents[68]. Thus, they are well placed to mediate coarser, large-amplitude stimuli, similar to our observation that early-born utricular afferents carry striolar information with inferred phasic responses. The mechanisms shaping topographic patterning in these two sensory systems may have been templates for

tonotopic organization in the auditory cochlear afferents that evolved later[69].

The link between developmental sequence and motor control has been shown most robustly in zebrafish[3–5,70] and fruit flies[6,7], suggesting that sequential developmental from fast to slow is an ancient principle of motor control. Building on these results, several classes of spinal neurons in mice have also been shown to differentiate by subtype according to their birth order[71,72]. However, whether these subtypes are tied to different speeds of movement remains to be explored.

Our work demonstrates that developmental sequence also links sensory inputs to motor outputs. The vestibular system is tightly coupled to motor control, and therefore it seems plausible that its organization relies on related principles. Our data show that the Mauthner cell preferentially receives utricular input from striolar-driven afferents (65% of utricular input). Notably, these afferents are the earliest-born vestibular pathways; similarly, the Mauthner cell drives the escape reflex as early as 2 dpf via the earliest-born spinal motor and premotor neurons[26]. Thus, the entire escape reflex arc runs from from early-developing hair cells, to early-developing afferents, to early-developing hindbrain reticulospinal neurons, to early-developing spinal elements and muscles. Both oculomotor and postural behaviors appear around 3–4 dpf in zebrafish[12,15,73–75], with continued maturation at later stages[14]. Correspondingly, the VS and VOR neurons develop after the Mauthner cell in zebrafish[38] (Goldblatt et al., 17th International Zebrafish Conference abstract, 2022), and these populations appear to develop at roughly the same time in amniotes[76]. In mouse, VS

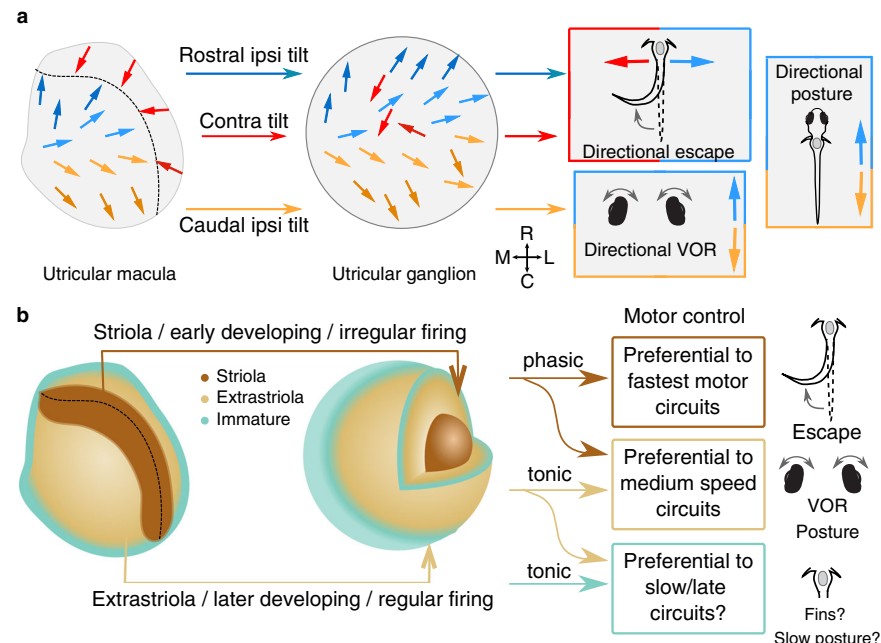

**Fig. 7 | Directional tuning and developmental sequence are organizing principles of vestibulomotor connectivity. a** Summary schematic of organization by directional tuning. Hair cells in the utricular macula, left, project via afferents that maintain rostrocaudal organization but not mediolateral organization. The utricular afferent ganglion is organized rostrocaudally, but contralateral tilt sensitive afferents are intermingled. These afferents project with different patterns to distinct brainstem targets, conferring directional sensitivity in the mediolateral (escape) or rostrocaudal (VOR, posture) pathways. Colors indicate directional tuning as previously. **b** Summary schematic of organization by temporal kinetics.

Early-born, striolar hair cells make synaptic connections to early-born afferents, whose cell bodies are positioned laterally in the utricular ganglion, and typically myelinated by the larval stage examined here. These early-born afferents, carrying phasic information about head movement, preferentially excite escape pathways, which consist of early-born, fast reticulospinal and spinal motor neurons and muscles. Postural and VOR reflex pathways rely more on the tonic and phasic-tonic signals arising from extrastriolar, slightly later-born pathways. We speculate that circuits carrying immature input, like $VS_{med}$, may project to motor circuits governing slower and more refined control of movement.

neurons preferentially synapse with slow motor neurons[77], consistent with our observation that they receive more extrastriolar and later-developing input than escape circuits do. It is plausible that developmental sequence may be significant in circuit assembly beyond the vestibulomotor pathways examined here.

We find that VS neurons can be subdivided based on axon trajectory, with differing proportions of striolar, extrastriolar, and immature input across groups. The fact that the $VS_{med}$ population, whose axons plunge directly towards the midline, get a higher proportion of utricular input from immature hair cells suggests that they might be later-born and participate in later-onset postural functions. Zebrafish refine their postural control over the pitch axis from 4 to 15 dpf by increasing use of fins and improved bout timing mediated in part by VS neurons[13,14,38]. We speculate that these $VS_{med}$ neurons may contribute to this refinement by preferentially connecting with fin motor neurons, which are located near the midline[78]. In mammals, the slower-onset portion of postural reflexes is mediated in part by a subset of lateral vestibular nucleus neurons ($LVN_C$) which collateralize to contact pontine reticulospinal neurons[79]. These similarities suggest that a VS population encoding tonic head movements could target multiple downstream targets for fine motor control. In contrast, the higher proportion of striolar inputs to $VS_{vent}$ suggests a function in rapid postural control. Together, our findings suggest a relationship within the VS population between the developmental time course of sensory innervation, axon trajectory, and the speed of behavioral responses.

Although our reimaging did not extend to the oculomotor and trochlear nuclei, the dorsal trajectory of caudal-tilt sensitive (nose-up) tangential neurons corresponds well with the dorsal position of inferior rectus and superior oblique (eyes-down) motor neurons in these areas[37]. Based on the earlier development of eyes-down motor

neurons[36], we hypothesize that tangential and SVN pathways contacting these motor neurons may be earlier born as well.

The directional tuning of afferent inputs to the Mauthner cell and the commissural escape neurons was opposite in direction. Thus each Mauthner cell receives two forms of utricular inputs tuned to ipsilateral tilt: monosynaptic innervation from ipsilateral vestibular afferents and disynaptic innervation from the contralateral side. The identity of these commissural escape neurons receiving utricular input is not known. Based on their anatomy, they are not similar to excitatory spiral fiber neurons, whose axons target the Mauthner axon cap[80], nor to inhibitory commissural neurons underlying left-right choice in the auditory system, whose axons target the Mauthner cell body[81]. Instead, these commissural utricular neurons appear to target the Mauthner ventral dendrite (Fig. 3c). Vestibular commissural neurons in goldfish have previously been interpreted as inhibitory, because their activation diminishes the amplitude of the antidromically triggered spike[82,83]. We speculate that these zebrafish commissural escape neurons are instead excitatory, because they carry signal that is similar to the predicted monosynaptic vestibular afferent inputs to that Mauthner cell. Alternatively, these commissural neurons might provide an inhibitory signal that sharpens lateralization, as described in the auditory and lateral line systems[81]. Further high-resolution imaging or physiology could resolve this question.

Additionally, because utricular inputs to the Mauthner cell and these commissural neurons are predominantly striolar, they are expected to be phasic and high-pass, with exquisite spike timing[45,84]. Irregular-firing afferents carrying striolar signals are thought to be kinetically well suited to signal translational movements, which are typically brief, in contrast to the lower-frequency movements generated by tilt[85]. In keeping with this prediction, we identified a vestibular-dependent escape behavior triggered by rapid translation. This

behavior may be important for survival: a predator approaching from the right would cause a fast translational head displacement to the left. This would activate both the ipsilateral tilt tuned afferents on the right side and the contralateral tilt tuned afferents on the left side of the animal (Fig. 3e). Via the direct and indirect pathways described here, these would summate to activate the right side Mauthner cell, which triggers contraction of the left side of the body, leading to a correctly directed escape away from the predator.

We find that VOR circuit neurons in the tangential and SVN all receive inputs tuned to ipsilateral tilt, with subpopulations aligned to either rostral or caudal tilt directions for control of vertical and torsional eye movements. The shared ipsilateral sensitivity explains why these subpopulations were not differentiated in zebrafish imaging studies of vestibular tuning in the roll axis[86,87]. Our anatomical data predict that VS neurons exhibit greater sensitivity to rostral tilt, consistent with our physiological results, in which VS neurons are preferentially sensitive to caudal translation[39]. Some VS neurons are contacted by divergently tuned afferents in the rostrocaudal axis and are predicted to have correspondingly more complex tuning responses, as seen in mammals[88,89]. These results align well with our observation that complex central tuning in VS neurons is constructed by feedforward excitation from afferents with differing tuning[39]. We also discovered that vestibular afferents innervating the most lateral portion of the macula (contralateral tilt sensitive) form a separate axon bundle in the hindbrain (Fig. 2d and Supplementary Movie 3). This result implies differential targeting of this pathway, consistent with the commissural utricular neurons in the escape circuit. The contralateral tilt-sensitive pathway is also thought to drive local feedforward inhibition to amplify target tuning by a push-pull mechanism[90].

Our observation that VOR and VS neurons receive similar mixtures of striolar and extrastriolar input are consistent with physiological analyses suggesting that both populations receive similar amounts of synaptic input from irregular and regular firing vestibular afferents[84,91,92]. In those analyses, irregular-firing (phasic) afferent inputs appeared to dominate, whereas in our anatomical analyses, striolar-driven afferents make up less than half the overall utricular input. We suggest that synaptic contacts onto VOR and VS neurons from irregular afferents are stronger in amplitude than those from regular-firing afferents, consistent with their morphologically larger synaptic contacts[93]. However, irregular-firing inputs preferentially encode high-frequency head movements, which are less common[94]. Therefore, their dominance in nerve stimulation experiments may reflect synaptic weights but not their overall influence on typical activity. Indeed, loss of striolar hair cells has little effect on oculomotor and basic postural behaviors[95], consistent with our observation that those pathways draw mostly on extrastriolar inputs. There are also likely to be significant variations across vertebrates in the balance between phasic and tonic vestibular signals, based on characteristics of movement[96].

Our analyses could not test whether developmental sequence *governs* connectivity, as opposed to the possibility that a genetically distinct class of early-born afferents preferentially connects to escape circuits. Future experiments manipulating circuit formation may shed light on this question. Our data also necessarily represent a single snapshot in time. Is the observed circuit, aligned with developmental sequence, likely to be maintained in adulthood? We suggest that it likely is, on the basis of the functional correspondence between early-born hair cells and early-born motor neurons which both display high-pass dynamics, while late-born hair cells and motor neurons display slower dynamics[97–99]. However, the spatial organization of the ganglion may become distorted by continued development; further experiments will be needed to address this question. It will also be important to evaluate how later-born hair cells in the striolar region integrate into the circuit. If they drive activity in later-born central circuits, their afferents may change central temporal dynamics. Alternatively, if they

integrate with early-born circuits as other striolar hair cells do, they may serve to augment existing signals.

## Methods

All experiments on animals were approved by the Washington University Institutional Animal Care and Use Committee.

Ultrathin (60 nm thick) serial sections from a 5.5 dpf larval zebrafish were a generous loan from J. Lichtman and F. Engert. Using the published $18.8 \times 18.8 \times 60$ nm³ per voxel and $56.4 \times 56.4 \times 60$ nm³ per voxel reference map and reconstructions[16], we re-imaged the right side of the fish, covering the utricular hair cells, utricular afferents (identified by their peripheral processes reaching the utricular macula), and a rostrocaudal extent of the brainstem that covered several major vestibular nuclei at $4.0 \times 4.0 \times 60$ nm³ per voxel. The volume covered ~100 μm in the rostrocaudal axis, 150 μm in the mediolateral axis, and 100 μm in the dorsoventral axis, in an irregular shape designed to capture the afferent peripheral and central processes (Fig. 1b). Imaging was carried out at the Washington University Center for Cellular Imaging on a Zeiss Merlin 540 FE-SEM with a solid-state backscatter detector. WaferMapper software[100] was used to control an ATLAS scan engine for automated focus and acquisition[101]. The resulting images were aligned onto the original $56.4 \times 56.4 \times 60$ nm³ per voxel dataset using affine and elastic transformations in FIJI's TrakEM2 plugin[102,103], with custom support from UniDesign Solutions.

The entire image volume was hosted in a CATMAID instance[104,105]. Vestibular circuit neurons and hair cells were reconstructed as skeletons, *i.e.* node points without volume fills. All utricular afferents were identified by stepping section by section through the entire anterior macula twice and marking every hair cell ribbon synapse. Ribbon synapses were identified by the characteristic large dark ribbon structure surrounded by vesicles (Fig. 2a). Every utricular afferent was followed as far as possible, in most cases to the corresponding soma in the vestibular ganglion. Only 1.6% (15/944) of processes adjacent to ribbons could not be followed to a soma due to the quality or ambiguity of the images.

Hair cell kinocilia and the tallest stereocilia were traced from the apical surface of each hair cell to their distal tips. The kinocilium was recognizable based on its distinctive structure (see Fig. 2a). Ciliary length was calculated as the sum of the Euclidean point-to-point distances. Positions of all cilia were plotted at the epithelial plane and a three-dimensional tuning vector for each hair cell was derived from the center of mass of all stereocilia to the kinocilium. Hair cell vector lengths were typically short in the dorsal-ventral axis relative to their extent in the other two axes (around one-third the normalized vector magnitude of the other two axes; Supplementary Fig. 1b), consistent with the mostly horizontal orientation of the utricular macula. Therefore, for the purposes of analysis we focused exclusively on their projection in the horizontal plane. Ganglion soma position was quantified in three dimensions for all analyses. During fixation, differential shrinkage caused a small tissue separation that is visible as a gap in the horizontal projection of the utricular ganglion reconstruction (Fig. 2g, upper right), but there was no loss of tissue. Tuning similarity was calculated as the cosine of the difference between the tuning directions of each pair of hair cells or afferents. Hair cell distance was determined by the 3D Euclidean distance between their kinocilia. Afferent distance was determined by the 3D Euclidean distance between their soma centers. Only hair cells on the medial side of LPR or afferents innervating the medial side of LPR were included for analysis of tuning similarity.

From the afferent somata in the utricular ganglion, afferent axons were then followed into the brain. A total of 105 afferents were successfully reconstructed by two experienced annotators (N.S. and M.W.B.) and all tracing was reviewed (M.W.B.). Central synapses were identified by close appositions, thickening of the presynaptic membrane, and clustered vesicles (e.g. Figs. 3a, 4a). The Mauthner cell was

previously reconstructed, as were most of the VS neurons[16,39]. Additional VS neurons were identified by reconstruction of utricular target neurons to the point that they joined up with previously traced myelinated axons. VOR neurons were identified based on their utricular input and their characteristic axonal projection patterns in the medial longitudinal fasciculus. Because much of the axonal projections lay outside the reimaged territory, only myelinated portions of axons could be reconstructed with confidence. Therefore, we were not able to follow some VOR axons all the way to the trochlear and oculomotor nuclei. Utricular commissural neurons were identified by their axons which crossed the midline and traveled to the contralateral Mauthner cell. We note that we have identified a large number of additional utricular target neurons in the brainstem that either do not fit into these categories (e.g., commissural neurons) or cannot be confidently identified due to the difficulty in extending their axons into lower-resolution territory. Therefore, the set of brainstem neurons analyzed here is likely to be strongly biased to early-born, or at least early-myelinated, and is not a complete description of all VS or VOR neurons. Nonetheless, the 19 VS neurons identified here is a large proportion of the 27 identified by retrograde labeling[38].

Behavioral data were acquired in 5–6 dpf larvae from wild-types or the line *rock solo*[AN66], an otogelin mutation[33]. Animals were visually verified as having normal (+/−) or absent (−/−) utricular otoliths. Animals were free-swimming in a small dish with infrared transillumination and imaged at 508 frames/s with a HiSpec-1 2 g monochrome camera mounted on a Scientifica SliceScope with a 2X objective. The translational stimulus was delivered with an air-floated sled (Aerotech ABL 1500WB) and designed for large amplitude acceleration and jerk, to maximize responses of irregular otolith afferents[106]. Due to the exploratory nature of this experiment, there was no attempt to separate larvae into responders and non-responders as has been done for acoustic stimuli[30], but animals were selected for behavioral tests based on whether they exhibited some response to dish tapping. Images were analyzed with ZebraZoom[107] version 1.28.12 (zebrazoom.org) to extract the smoothed tail angle and heading direction.

Synaptic clustering was measured as the average distances between synapses on the same dendritic arbor. Vestibular neuron arbor skeletons were first simplified for analysis by regularizing the distance between nodes to ~1 um. Each synapse was associated with the nearest skeleton node, and distances between synapses were calculated along the Euclidean length of the skeleton.

To understand how the observed mean distance between synapses compared to random distributions of synapses on the vestibular neuron arbors, we used several Monte Carlo models. For each model, the synapses of each vestibular neuron were redistributed across the arbor 100,000 times. The mean distance between synapses was measured the same way as the non-randomized mean distances was measured. The measure used for each of the 100,000 iterations was the average of the mean synaptic distances of 43 vestibular neurons observed for that iteration. Confidence intervals were calculated as the bounds containing 95% of the results of the 100,000 experimental randomizations.

The first model (unweighted distribution) compares the observed clustering of synapses to what would be expected if synapses of each vestibular neuron were randomly distributed across the entire arbor of that neuron. In this model, each vestibular neuron skeleton node had an equal probability of being randomly assigned to one of the synapses of that vestibular neuron.

In the second model (50 μm-from-afferent weighted distribution) we restricted nodes likely to be assigned to a synapse using the proximity of the nodes to the synaptic terminals of afferent axons. 105 afferent axons were skeletonized as above. We then counted the number of afferent synapse nodes within 50 μm of each vestibular neuron skeleton node (proximity score), subtracting 1 to remove the influence of synaptic afferents presynaptic to a given vestibular neuron

node. We next shuffled synaptic locations while maintaining the likelihood of synaptic connectivity given this proximity score. For example, if a skeleton node with a score of 95 has a 7% chance of being a synaptic locus, whereas a skeleton node with a score of 60 has a 1% chance, then nodes with scores of 95 will be given a 7-fold greater chance of being assigned a synapse in the redistribution than nodes with scores of 60.

In the final, most restrictive model (<5 μm-from-afferent weighted distribution), we attempt to use proximity to afferent terminals to recapitulate the clustering we observe in the real data. Proximity scores were calculated as +1 for each vestibular neuron node 0 μm from a presynaptic node. The added score decreased linearly to 0 at 5 μm distance. The rest of the 5 μm model was the same as the 50 μm model.

Analyses and statistics were carried out in Igor Pro (Wavemetrics) or Matlab (Mathworks). Statistical tests were carried out as reported in text, two-tailed where relevant, and typically with nonparametric analyses due to the non-normal distribution of parameters.

### Reporting summary
Further information on research design is available in the Nature Research Reporting Summary linked to this article.

## Data availability
The electron micrograph image data generated in this study is hosted online and publicly available at http://zebrafish.link/hildebrand16/data/vestibular_right. Source Data are provided with this paper. Within the Source Data, one folder contains the .json files needed to generate all 3D reconstructions plus instructions for use; an Excel (.xlsx) file contains data for most of the figure panels, as well as links to the online source of example images, and another folder contains central neuron reconstructions used in dendritic analyses in Fig. 5. Tabular quantification of hair cell to afferent and afferent to central target connectivity are provided (Supplementary Data 1, 2). Source data are provided with this paper.

## Code availability
Code used in generating the Monte Carlo analyses and sensory tuning analyses are available at https://github.com/bagnall-lab/EM_analysis

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

## Acknowledgements

We thank Drs. Mohini Sengupta, Vamsi Daliparthi and Mr. Saul Bello Rojas for thoughtful critiques of the paper. We are grateful to Drs. Daniel Kerschensteiner, David Schoppik and members from the Schoppik lab for insightful comments. Serial-section EM imaging was performed in part through the use of Washington University Center for Cellular Imaging (WUCCI) supported by Washington University School of Medicine, The Children's Discovery Institute of Washington University and St. Louis Children's Hospital (CDI-CORE-2015-505 and CDI-CORE-2019-813) and the Foundation for Barnes-Jewish Hospital (3770 and 4642). This work is supported by funding through the National Institutes of Health (NIH) R01 DC016413 (M.W.B.), EY030623 (J.L.M.), EY029313 (J.L.M.), a Sloan Research Fellowship (M.W.B.), a Fellowship in Neuroscience from the Leon Levy Foundation and a NARSAD Young Investigator Grant from the Brain and Behavior Research Foundation (D.G.C.H.), an unrestricted grant to the Department of Ophthalmology and Visual Sciences from Research to Prevent Blindness (J.L.M.), and a Research to Prevent Blindness Career Development Award (J.L.M.). M.W.B. is a Pew Biomedical Scholar and a McKnight Foundation Scholar.

## Author contributions

Z.L., Y.J., and M.W.B. conceived the project, performed the experiments and analyzed the data. J.L.M. and D.G.C.H. carried out the alignment of the serial-section EM image data. J.L.M. designed and carried out the Monte Carlo simulations. Z.L., M.W.B., Y.J., and N.S. completed the reconstructions. Z.L. and M.W.B. wrote the manuscript with input from all other authors.

## Competing interests

The authors declare no competing interests.
