## [Peer Review File · Nature Communications]

Organization of the gravity-sensing system in zebrafishREVIEWER COMMENTS

Reviewer #1 (Remarks to the Author):

Liu et al present a synaptic connectivity analysis of inner ear hair cells and utricular ganglion afferents of a larval zebrafish. They identify a spatial pattern and connectivity that correlates inferred hair cell development sequence with the onset of motor behaviors. Overall the study is well presented and the data support the conclusions of the authors.

Major comments:

1. The reconstruction approach for hair cells and the utricular ganglion is quite clear. However, the connectivity with hindbrain circuits less so. How exactly were the postsynaptic utricular commissural neurons, Tan, SVN and VS neurons found and identified? It's not sufficiently explained in the text, and that leaves open the possibility of a biased sampling of these populations.
2. A summary table of how many afferents targeted each brain stem region would help. As far as I can tell: 25 to Mauthner cell system, 35 to tangential VN, 29 to SVN, and 61 to VS neurons. That sums to 150 afferents, so at least some afferents are branching and diverging to more than one target nucleus? If this is correct, have the authors examined whether there is a pattern in which afferents diverge to multiple target nuclei?
3. The classification of SVN and Tan neurons into "blue" and "orange" based on axonal trajectory seems arbitrary, since the differences in axonal trajectory are rather slight. I suggest to instead map a continuous medio-lateral or dorso-ventral position variable onto the hair cells. Seeing a clear reflection of that in the hair cell tuning would be much more convincing.
4. The final panel of Fig 7d appears slightly too simplified as diagrammed. The weighted inputs to the VOR system appear nearly equally divided between S and ES contacting afferents in Fig 7a, but the summary suggests primarily tonic inputs are driving the VOR system.
5. Most figures containing reconstructions are missing scale bars. Please add.

Minor comments:

1. Line 184: mentions 105 afferents, figure legend indicates 104
2. Line 131: Are the centers of mass measured at the bases of the stereocilia? Please clarify (also in Methods section)
3. Line 229: Should reference Fig 3d?
4. Figure 3 legend, description of panels c and d are inverted

5. Line 313: What does 'phase computations' mean?
6. Line 339: 'and therefore inhibit...', is this a hypothesis or has been demonstrated?
7. Line 348: 'Notably...', is this/could this be illustrated in a figure?
8. Line 370: perhaps better phrasing, i.e., the synapses were clustered on dendrites
9. Line 424: what does 'rapid transformation into motor outputs' mean? What about this organization makes it rapid?
10. Line 490: 'irregular afferents', do you mean irregular-firing afferents?
11. Line 511: Suppl Fig 4a not previously cited in results. Should this point be seen in Fig 3a?
12. Line 513-514: Unclear statement, please reword or expand
13. Line 600: Does the reconstruction of only myelinated VOR axons bias the results to a subset of earlier-born neurons?
14. Figure legend 7a references Fig. 3e, this seems incorrect.

Reviewer #2 (Remarks to the Author):

This is a review of "The organization of the gravity-sensing system in zebrafish". In this reviewer's opinion, this manuscript was a pleasure to read and I recommend it for publication.

In this study, Liu and colleagues used high-resolution serial section electron microscopy to characterize the anatomical inputs and outputs of Zebrafish utricular afferents. The authors find two-dimensional topographical organization of the system. Along the rostral-caudal axis there is a map of (anatomically inferred) directional tuning for both the afferents in the utricular ganglion and the utricular hair cells in the utricular macula. Along the medial-lateral axis, organization is linked to developmental sequence, with early-born utricular afferents located medially and later-born afferents laterally. In the input side, a similar birth ordered sequence is found in the macula. On the output side, early-born afferents tend to synapse with neurons governing fast escape circuits, whereas later-born afferents tend to synapse with neurons of the slower VOR circuit.

This study is largely based on a straight-forward anatomical dataset. The approaches that they use are the gold-standard for the field, and therefore appropriate. Further, the data have been appropriately analyzed with supporting quantifications.

This study provides a significant advance in our understanding of motor circuits. It is becoming increasingly well accepted that a fundamental aspect of motor system organization is the temporal assembly. Specifically, early-born motor neurons trigger fast motor outputs like escape, and later-born neurons are used for more refined movement. This study links to and extends these concepts by looking at how input stimuli that regulate motor output may or may not obey similar principles. The authors' data suggest that temporal assembly may extend to sensory systems as well.

The expertise of this review is in development of motor systems and connectomics, but not Zebrafish nor gravity-sensing systems. So, I refrain from comments pertaining to those subjects.

Below is a list of minor comments that would increase the readability of the manuscript:

Throughout, including in the abstract, it would be useful to modify the use of the word "tuning" to something like "anatomically-inferred tuning". As it was very confusing at first, because tuning brings to mind a physiological measurement.

Providing a bit more information about the biology of a 5.5 dpf Zebrafish in the introduction and or first mention in the results section would greatly enhance the readability for non-Zebrafish audience.

In general, Figure 3 and the associated text was the most difficult to follow.

Lines 202-204 was confusing.

Lines 212-221 were confusing.

Figure 3c. Perhaps some transparency on these arrows to make overlaps more visible.

Is the Figure legend for 3c and d reversed?

RE Lines 246-247. The reference here isn't clear to me -- to some other work (already cited?) or to earlier text in this paper?

I was a bit bothered by the escape behavior experiment. I'd like more detail about the stimuli provided (relationship bw stim, initial animal orientation, & response occurring/in what direction), and I feel like it's missing a negative control. Specifically

RE: Line 309 “ directionally-biased”. What about the translational stimuli applied? Were they timed in some way to always translate animals along their medial-lateral axis? Were larvae biased to one side even if translation was aligned with their rostral-caudal axis instead? (I might want a plot of translation angle against successful escape angles, or more details in Methods about stimulus delivery & how it was compared to direction of escape).

RE: Line 313: “allows for phase computations to identify the direction”. This phrase makes me definitely want more info about orientation of translational stim vs. escape direction/failure to escape, in light of this hypothesis. Further, a comment in the Discussion later (490-492) got me thinking that a negative control is important for this claim too... is it only the high-frequency vestibular stimuli that elicit this escape, or could slower ones do it too?

RE: Figure 5

It took me a while reading this (before the text) to figure out that green synapses were from same-side utricular afferents, & not from red 'commissural utricular' neurons -- might want to add representation of 1ry ganglion, or green synapses on c) as well?

RE:Lines 368-370.

It is unclear to me from the figure & the methods what this distribution metric corresponds to. How does 'concentration of dendritic nodes' relate to the null model of synapse distribution?

Reviewer #3 (Remarks to the Author):

In this manuscript, Zhikai Liu and colleagues report their work on the organisation of the utricle of the vestibular system in young zebrafish. They use a dataset of serial-section electron microscopy to trace neurons, and observe connectivity with hair cells and central targets. The work is exhaustive and very well conducted. The quality of the data is exceptional and their results are sound and very well presented. I do not have anything to add to make the manuscript better. My very important concern is that this work is purely descriptive. What we learn from this work is rather limited. I do like good descriptions and feel that they serve a purpose, but I believe that they should be published in journals that serve that specific purpose.

We are grateful for the thoughtful review comments on our manuscript and have made every effort to address them thoroughly in this revision. Below are point-by-point responses, but in summary our major changes are:

- 1) In response to concerns about the clarity of the presentation with respect to directional tuning, we have significantly revamped the figures. We now treat inferred directional tuning and temporal dynamics in separate figures, rather than trying to address them simultaneously, and we have added a directional color code that is consistent throughout the paper. This necessitated some figure rearrangement, but we hope that the revised manuscript will be clearer and more accessible to all readers.
- 2) We have carried out a more extensive behavioral analysis, now detailed in Fig. 3, of the vestibular-evoked escape response. In particular, we added a unidirectional translational stimulus and found that it evoked highly consistent behavioral responses with respect to the animal's heading direction (Fig. 3g). This result matches well with the directional escape circuit we identified.
- 3) We re-analyzed the concentration of synapses on vestibular neuron dendrites by measuring the distances between synapses along the length of the neuron arbors. As a comparison, we used three Monte Carlo models to generate a set of random distributions of synapses with different anatomical constraints (Fig. 4e). We found that the observed distance is in line with the distribution of the most restrictive model, and much shorter than those with less or no anatomical constraint. This result indicates that the observed synapses are more concentrated on the postsynaptic arbor than expected, and the anatomical proximity between the afferents and postsynaptic targets likely plays an important role in shaping this synaptic distribution.

In addition, we have addressed all reviewer comments, as detailed below (reviewer comments in italics).

Reviewer #1 (Remarks to the Author):

Liu et al present a synaptic connectivity analysis of inner ear hair cells and utricular ganglion afferents of a larval zebrafish. They identify a spatial pattern and connectivity that correlates inferred hair cell development sequence with the onset of motor behaviors. Overall the study is well presented and the data support the conclusions of the authors.

Major comments:

1. The reconstruction approach for hair cells and the utricular ganglion is quite clear. However, the connectivity with hindbrain circuits less so. How exactly were the postsynaptic utricular commissural neurons, Tan, SVN and VS neurons found and identified? It's not sufficiently explained in the text, and that leaves open the possibility of a biased sampling of these populations.

The reviewer is correct that the successful identification of these neurons was biased towards neurons with sufficiently myelinated axons to permit tracing outside the re-imaged volume. We agree that this introduces bias, and we think that the bias is, in effect, to early-born neurons (as in minor comment 13 below). We have addressed this limitation much more explicitly in the

Results and Methods. Specifically, we added descriptions of the identification process and associated bias [Lines 140-148]. However, we think that this bias does not affect our conclusions substantially because it applies equally well across the utricular target populations.

In the course of addressing this question, we re-scanned the images for any potential VS and VOR neurons missed in the original round of reconstructions and found several that are now included in the revised manuscript (see especially Fig. 6 with 4 new VS neurons). We are also publishing a separate manuscript that describes the remaining unidentified utricular target neurons, where either the axon could not be found or it could not be traced far enough to establish identity robustly (Jia and Bagnall, submitted to *Frontiers*). That work complements the existing manuscript and provides more thorough accounting of the unidentified utricular targets.

2. A summary table of how many afferents targeted each brain stem region would help. As far as I can tell: 25 to Mauthner cell system, 35 to tangential VN, 29 to SVN, and 61 to VS neurons. That sums to 150 afferents, so at least some afferents are branching and diverging to more than one target nucleus? If this is correct, have the authors examined whether there is a pattern in which afferents diverge to multiple target nuclei?

Some afferents indeed diverge to multiple target nuclei. We have included a Supplemental Table 2 that provides the entire matrix of connectivity, as well as connectivity grouped by target nucleus, to show the divergence patterns. We found that any two target nuclei receive common inputs, and there didn't seem to be any clear pattern with respect to targets. However, myelinated afferents do target more nuclei than unmyelinated afferents (9/16 myelinated diverge to all four, 3/16 diverge to three nuclei, and 1/16 diverge to two nuclei). In comparison, 0/90 unmyelinated afferents diverge to all four nuclei, 9/90 diverge to three, and 25/90 innervate two. We think this is consistent with the idea that myelinated afferents are earlier-born and more mature, and have included this observation in the text [Lines 345-348].

3. The classification of SVN and Tan neurons into "blue" and "orange" based on axonal trajectory seems arbitrary, since the differences in axonal trajectory are rather slight. I suggest to instead map a continuous medio-lateral or dorso-ventral position variable onto the hair cells. Seeing a clear reflection of that in the hair cell tuning would be much more convincing.

This was a very helpful suggestion. We couldn't map back to hair cells because some afferents were shared across the groups, but instead this idea prompted us to create a coherent color scheme for directional tuning, which is now implemented in Figs. 2-4. In particular, we plot axon position for these groups and colorize by computed directional tuning in Fig. 4, allowing readers to see the relationships more clearly.

4. The final panel of Fig 7d appears slightly too simplified as diagrammed. The weighted inputs to the VOR system appear nearly equally divided between S and ES contacting afferents in Fig 7a, but the summary suggests primarily tonic inputs are driving the VOR system.

Agreed and we have revamped the summary schematic accordingly.

5. Most figures containing reconstructions are missing scale bars. Please add.

Apologies for the oversight; this has now been fixed.

We have corrected or addressed all minor comments. In particular for #7, this is now illustrated in Fig. 3g; and for #11, this is now in Fig. 2d as well.

Minor comments:

1. Line 184: mentions 105 afferents, figure legend indicates 104
2. Line 131: Are the centers of mass measured at the bases of the stereocilia? Please clarify (also in Methods section)
3. Line 229: Should reference Fig 3d?
4. Figure 3 legend, description of panels c and d are inverted
5. Line 313: What does 'phase computations' mean?
6. Line 339: 'and therefore inhibit...', is this a hypothesis or has been demonstrated?
7. Line 348: 'Notably...', is this/could this be illustrated in a figure?
8. Line 370: perhaps better phrasing, i.e., the synapses were clustered on dendrites
9. Line 424: what does 'rapid transformation into motor outputs' mean? What about this organization makes it rapid?
10. Line 490: 'irregular afferents', do you mean irregular-firing afferents?
11. Line 511: Suppl Fig 4a not previously cited in results. Should this point be seen in Fig 3a?
12. Line 513-514: Unclear statement, please reword or expand
13. Line 600: Does the reconstruction of only myelinated VOR axons bias the results to a subset of earlier-born neurons?
14. Figure legend 7a references Fig. 3e, this seems incorrect.

Reviewer #2 (Remarks to the Author):

This is a review of "The organization of the gravity-sensing system in zebrafish". In this reviewer's opinion, this manuscript was a pleasure to read and I recommend it for publication.

In this study, Liu and colleagues used high-resolution serial section electron microscopy to characterize the anatomical inputs and outputs of Zebrafish utricular afferents. The authors find two-dimensional topographical organization of the system. Along the rostro-caudal axis there is a map of (anatomically inferred) directional tuning for both the afferents in the utricular ganglion and the utricular hair cells in the utricular macula. Along the media-lateral axis, organization is linked to developmental sequence, with early-born utricular afferents located medially and later-born afferents laterally. In the input side, a similar birth ordered sequence is found in the macula. On the output side, early-born afferents tend to synapse with neurons governing fast escape circuits, whereas later-born afferents tend to synapse with neurons of the slower VOR circuit.

This study is largely based on a straight-forward anatomical dataset. The approaches that they use are the gold-standard for the field, and therefore appropriate. Further, the data have been appropriately analyzed with supporting quantifications.

This study provides a significant advance in our understanding of motor circuits. It is becoming increasingly well accepted that a fundamental aspect of motor system organization is the temporal assembly. Specifically, early-born motor neurons trigger fast motor outputs like escape, and later-born neurons are used for more refined movement. This study links to and extends these concepts by looking at how input stimuli that regulate motor output may or may not obey similar principles. The authors' data suggest that temporal assembly may extend to sensory systems as well.

The expertise of this review is in development of motor systems and connectomics, but not Zebrafish nor gravity-sensing systems. So, I refrain from comments pertaining to those subjects.

Below is a list of minor comments that would increase the readability of the manuscript:

Throughout, including in the abstract, it would be useful to modify the use of the word "tuning" to something like "anatomically-inferred tuning". As it was very confusing at first, because tuning brings to mind a physiological measurement.

This is a very good point. We have included more references to "inferred" tuning but we have also addressed this issue directly in the Results [Lines 118-121] to acknowledge the limitations of using EM to calculate tuning.

Providing a bit more information about the biology of a 5.5 dpf Zebrafish in the introduction and or first mention in the results section would greatly enhance the readability for non-Zebrafish audience.

Agreed and we have added this to the Introduction [Lines 44-50].

In general, Figure 3 and the associated text was the most difficult to follow.

Lines 202-204 was confusing.

Lines 212-221 were confusing.

Figure 3c. Perhaps some transparency on these arrows to make overlaps more visible.

Is the Figure legend for 3c and d reversed?

We have completely overhauled this section in response to this and other comments. We agree that trying to evaluate directional tuning by vectors was a bit much, and accordingly have implemented a color code for clarity of reading. In addition, we separated the sections on inferred directional tuning and development/temporal dynamics. We hope that this substantial revision improves the clarity.

RE Lines 246-247. The reference here isn't clear to me -- to some other work (already cited?) or to earlier text in this paper?

We have clarified with citations to the developmental growth of the vestibular ganglion.

I was a bit bothered by the escape behavior experiment. I'd like more detail about the stimuli provided (relationship bw stim, initial animal orientation, & response occurring/in what

direction), and I feel like it's missing a negative control. Specifically

RE: Line 309 “ directionally-biased”. What about the translational stimuli applied? Were they timed in some way to always translate animals along their medial-lateral axis? Were larvae biased to one side even if translation was aligned with their rostral-caudal axis instead? (I might want a plot of translation angle against successful escape angles, or more details in Methods about stimulus delivery & how it was compared to direction of escape).

To address this we added new behavioral experiments. Our original translational stimulus involved delivering a slow movement in one direction and a rapid jerk back in the other direction to achieve a sufficiently precisely timed stimulus that we could assess the latency of the C-bend, as latency is the main criterion for showing Mauthner cell involvement (Fig. 3f). However, the bidirectional nature made it less optimal for evaluating directionality. Therefore, we added a new unidirectional stimulus, again in free-swimming animals, to evaluate the directionality of the response. Indeed, in this scenario 95% of escapes occurred in the direction predicted by the inferred directional tuning of the Mauthner circuit! And as the reviewer predicted, escapes in the “incorrect” direction occurred when the animal was being accelerated in a mostly rostral or caudal direction. We quantify this new result in Fig. 3g. Thank you for the suggestion.

RE: Line 313: “allows for phase computations to identify the direction”. This phrase makes me I definitely want more info about orientation of translational stim vs. escape direction/failure to escape, in light of this hypothesis. Further, a comment in the Discussion later (490-492) got me thinking that a negative control is important for this claim too... is it only the high-frequency vestibular stimuli that elicit this escape, or could slower ones do it too?

We also carried out some additional experiments to try to assess whether escape is best elicited by high-frequency stimuli. There were some experimental limitations imposed by our rig even with modifications, in that we could not accelerate beyond a certain limit. However, within the parameter space we were able to try, we did not find any support for the idea that high-frequency stimuli are more successful at eliciting escapes. We have therefore removed this claim.

RE: Figure 5

It took me a while reading this (before the text) to figure out that green synapses were from same-side utricular afferents, & not from red 'commissural utricular' neurons -- might want to add representation of 1ry ganglion, or green synapses on c) as well?

This figure has been recolored to align with directional tuning, hopefully clarifying this display issue.

RE: Lines 368-370.

It is unclear to me from the figure & the methods what this distribution metric corresponds to. How does 'concentration of dendritic nodes' relate to the null model of synapse distribution?

Originally we had conceived the quantification of synapse clustering in a circular framework, where we treated each dendrogram as a radial distribution and then compared the concentration of synapses relative to the concentration of “nodes” (i.e. points along the reconstructed skeleton).

However, the reviewer's comment highlighted that this is not a particularly intuitive or common analysis, and therefore we completely redid this analysis. In the new Fig. 4e, we compare the actual synaptic clustering (distance between synapses) with three Monte Carlo simulations under different anatomical constraints: random distribution, distribution within 50 μm , and distribution within 5 μm of afferents. The results support the observation that synaptic inputs are clustered, within constraints applied by the relative position of the afferents and the central neurons.

Reviewer #3 (Remarks to the Author):

In this manuscript, Zhikai Liu and colleagues report their work on the organisation of the utricle of the vestibular system in young zebrafish. They use a dataset of serial-section electron microscopy to trace neurons, and observe connectivity with hair cells and central targets. The work is exhaustive and very well conducted. The quality of the data is exceptional and their results are sound and very well presented. I do not have anything to add to make the manuscript better. My very important concern is that this work is purely descriptive. What we learn from this work is rather limited. I do like good descriptions and feel that they serve a purpose, but I believe that they should be published in journals that serve that specific purpose.

We appreciate the reviewer's kind words about the quality of the work. With respect to the descriptive aspect of this manuscript: first, the behavioral result of directional escapes in the new Figure 3 is a nice test of the predictions derived from circuitry, suggesting the utility of the data and providing a distinctly experimental approach. Second, we agree that the rest of this work is largely descriptive. Many of the newer "big data" techniques in neuroscience, like single-cell RNAseq, are currently producing large quantities of descriptive data that eventually will become integrated into more experimental manipulations. We think that our work is valuable both in setting a baseline for future work, and because we specifically identify some topographical relationships in the vestibular system that have not been shown previously. As topography is a classic and fundamental approach to understanding sensorimotor circuits, we think this result stands on its own.

REVIEWERS' COMMENTS

Reviewer #1 (Remarks to the Author):

The authors did a great job addressing my questions and suggestions. I recommend the manuscript for publication

Reviewer #2 (Remarks to the Author):

My concerns have been address.